# Neuron populations use variable combinations of short-term feedback mechanisms to stabilize firing rate

**Sarah Pellizzari**[1], **Min Hu**[1], **Lara Amaral-Silva**[2], **Sandy E. Saunders**[2], **Joseph M. Santin**[2]*

1 University of North Carolina at Greensboro, Greensboro, North Carolina, United States of America,
2 University of Missouri, Columbia, Missouri, United States of America

* santinj@missouri.edu

**Data Availability Statement:** All relevant data are within the paper and its Supporting information files.

## Abstract

Neurons tightly regulate firing rate and a failure to do so leads to multiple neurological disorders. Therefore, a fundamental question in neuroscience is how neurons produce reliable activity patterns for decades to generate behavior. Neurons have built-in feedback mechanisms that allow them to monitor their output and rapidly stabilize firing rate. Most work emphasizes the role of a dominant feedback system within a neuronal population for the control of moment-to-moment firing. In contrast, we find that respiratory motoneurons use 2 activity-dependent controllers in unique combinations across cells, dynamic activation of an $Na^+$ pump subtype, and rapid potentiation of Kv7 channels. Both systems constrain firing rate by reducing excitability for up to a minute after a burst of action potentials but are recruited by different cellular signals associated with activity, increased intracellular $Na^+$ (the $Na^+$ pump), and membrane depolarization (Kv7 channels). Individual neurons do not simply contain equal amounts of each system. Rather, neurons under strong control of the $Na^+$ pump are weakly regulated by Kv7 enhancement and vice versa along a continuum. Thus, each motoneuron maintains its characteristic firing rate through a unique combination of the $Na^+$ pump and Kv7 channels, which are dynamically regulated by distinct feedback signals. These results reveal a new organizing strategy for stable circuit output involving multiple fast activity sensors scaled inversely across a neuronal population.

## Introduction

Neural circuits must produce stable output in an ever-changing environment or else a suite of neurological disorders follow [1,2]. To accomplish this goal, many neurons can track their firing rate in real time and make rapid physiological adjustments to stabilize activity. One important strategy involves cellular systems that transduce firing rate through the dynamics of $Ca^{2+}$, $Na^+$, or voltage, which then alter excitability over the next several seconds to keep activity from drifting into an unhealthy range [3–9]. These mechanisms rely on fast negative feedback and have a potent stabilizing effect on neuronal output. However, they differ from what is often termed "homeostatic plasticity," as intrinsic biochemical and physical dynamics maintain firing rate within a tight range rather than bringing activity back into line following a large

**Funding:** This work was funded by the National Institutes of Health (R01NS114514); https://www.nih.gov/) to JS and the U.S. Department of Defense (76129-RT-REP; https://www.defense.gov/) to JS. The funders had no role in study design, data collection and analysis, decision to publish, or preparation of the manuscript.

**Competing interests:** The authors have declared that no competing interests exist.

perturbation [10–12]. Although many signals feed back onto neurons, a design feature of negative feedback systems across biology involves a dominant sensory signal for a given regulated variable, e.g., changes in pressure for yeast osmoregulation [13] and pH for blood gas regulation [14]. This organization may prevent the unstable output that occurs when multiple signals with different set points are used to maintain homeostasis of a single process [15,16]. Thus, most studies on the moment-to-moment regulation of neuronal output address how a key feedback signal controls firing rate within a given cell type [3–9].

The $Na^+/K^+$ ATPase is best known for its housekeeping role in ion regulation; however, it is expressed selectively in some neurons to control excitability through activity-dependent feedback [17]. After a burst of action potentials, the loading of intracellular $Na^+$ recruits a subtype of the $Na^+/K^+$ ATPase with a low affinity for $Na^+$ (referred to here as the "dynamic" $Na^+$ pump). When this pump kicks on, it hyperpolarizes the membrane for tens of seconds to reduce the probability of future firing [18]. Feedback control by the dynamic $Na^+$ pump plays an important role in animal behavior, as it shapes locomotor performance in insects [19], amphibians [3,20], and rodents [21]. Therefore, the dynamic form of the $Na^+$ pump represents a conserved sensor of $Na^+$ dynamics, which controls neuronal output in response to recent firing.

We have been investigating mechanisms that underly reliable output of the respiratory network in amphibians [22–24]. Given the role of the dynamic $Na^+$ pump in locomotion [3,20], we tested its role in respiratory motoneurons. In beginning this work, we identified activity-dependent feedback characteristic of the dynamic $Na^+$ pump, with intense stimulation leading to membrane hyperpolarization over the following minute. Because the dynamic $Na^+$ pump generates outward current without opening an ion channel, this "ultraslow afterhyperpolarization" (usAHP) does not usually occur with changes in membrane input resistance ($R_{in}$) [3,21]. Thus, we were struck as this usAHP was accompanied by a range of decreases in $R_{in}$ from cell to cell, spanning from small to large. This result led us to hypothesize that variable combinations of activity-dependent mechanisms involving the dynamic $Na^+$ pump and fast enhancement of a $K^+$ channel, which would decrease $R_{in}$, stabilize activity across this population of motoneurons. Below, we detail how 2 feedback controllers, the dynamic $Na^+$ pump (a $Na^+$ sensor) and Kv7 channels (through voltage-sensitive potentiation), scale reciprocally across neurons for the fast control of firing rate. These results indicate that stable circuit output can arise through unique sets of rapid activity sensors spread throughout a neuronal population.

## Results

### Spike-dependent and independent feedback cause the usAHP

We began this study to assess the role of feedback from the dynamic $Na^+$ pump in motoneurons that gate lung airflow in American bullfrogs (vagal motoneurons; Fig 1A). In locomotor neurons, $Na^+$ loading during burst firing activates the pump and triggers an usAHP that lasts for approximately 30 to 60 seconds to homeostatically reduce membrane excitability [3,18,20]. Respiratory motoneurons exhibited a similar usAHP (Fig 1A). However, we found that hyperpolarization corresponded with decreases in $R_{in}$ that recovered with the membrane potential following stimulation. $R_{in}$ changes were variable from cell to cell, spanning from roughly no change to a 40% drop (Fig 1B and 1C, and S1 Fig). The slope of the usAHP did not correlate with the reduction in $R_{in}$ (Fig 1D), suggesting that multiple mechanisms combine to generate a phenotypically similar usAHP across cells. Indeed, neurons with stable $R_{in}$ following stimulation had an usAHP triggered by spiking, as the block of action potentials during stimulation reduced the usAHP (Fig 1E and 1F). In contrast, neurons with relatively large decreases in $R_{in}$

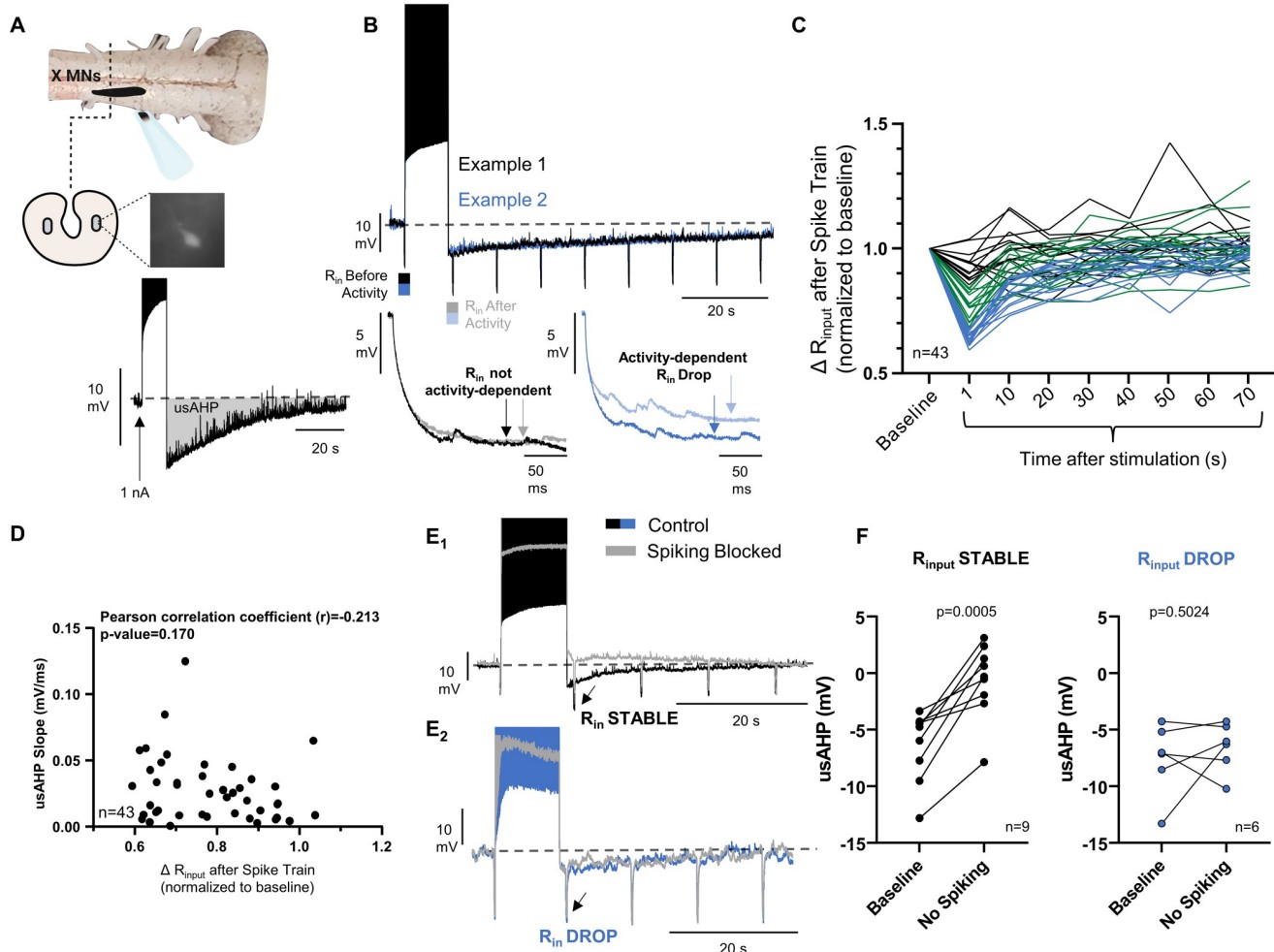

**Fig 1. The usAHP involves multiple mechanisms driven by different activity signals.** (A) Identified vagal motoneurons that innervate the glottal dilator exhibit short-term regulation of excitability following intense firing, termed the usAHP. (B) The usAHP is associated with variable $R_{in}$ changes across neurons, with 1 example showing largely stable $R_{in}$ (black/gray) and another with a decrease (blue). (C) $R_{in}$ during the usAHP reported as a relative change from baseline in $n = 43$ neurons following stimulation, color coded by no (black), modest (green), and large (blue) $R_{in}$ changes. (D) The usAHP does not correlate with size of the $R_{in}$ drop ($n = 43$ neurons from $N = 15$ animals). (E) Different activity signals drive the usAHP in association with the change in $R_{in}$. Neurons with stable $R_{in}$ have an usAHP driven by spiking, and usAHPs with the largest $R_{in}$ decreases persist without spiking. (F) Mean data showing the influence of spiking on the usAHP (paired $t$ test). Stable $R_{in}$ ($n = 9$ neurons from $N = 2$ animals) and $R_{in}$ decreases ($n = 6$ neurons from $N = 6$ animals). The data underlying this figure can be found in S1 Data.

retained the usAHP during stimulation without spiking (Fig 1E and 1F). Therefore, combinations of 2 feedback mechanisms, with spike-dependent and independent components, regulate membrane excitability over the same timescale of about 1 minute.

## Scaling of the dynamic Na⁺ pump and Kv7 channels give rise to the usAHP

A spike-sensitive usAHP with stable $R_{in}$ is consistent with feedback from the dynamic $Na^+$ pump [3]. However, reduced $R_{in}$ during the usAHP pointed to the activation of a $K^+$ conductance. At least 2 $K^+$ channels have been shown to be enhanced by short depolarization without spiking over the time course we observe here (roughly 1 minute): $Ca^{2+}$-activation of $K^+$ leak channels [9] and depolarization-induced potentiation of Kv7 channels [4]. Therefore, we tested whether inhibitors of the dynamic form of the $Na^+$ pump (2 μM ouabain), $Ca^{2+}$

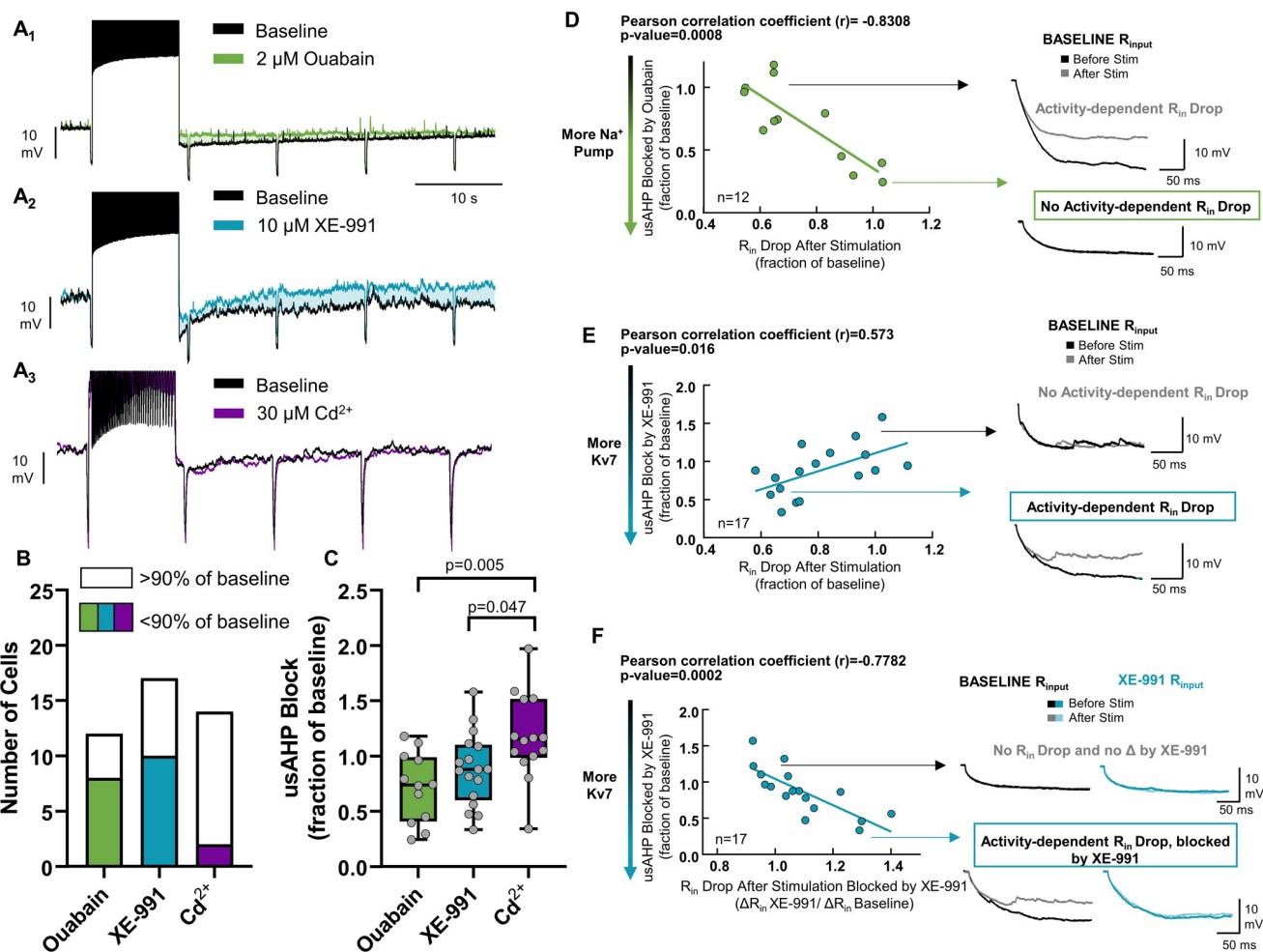

**Fig 2. The dynamic Na$^+$ pump and Kv7 channels account for short-term feedback, with variable contributions from cell to cell.** (A) Example recordings showing the effect of 2 μM ouabain (dynamic Na$^+$ pump, $n = 12$ cells from $N = 8$ animals), 10 μM XE-991 (Kv7, $n = 17$ cells from $N = 5$ animals), and 30 μM Cd$^{2+}$ (Ca$^{2+}$ channels, $n = 14$ neurons from $N = 4$ animals) on the usAHP. (B) 2 μM ouabain and 10 μM XE-991 reduced the usAHP in most neurons, while 30 μM Cd$^{2+}$ did not. The frequency of positive effects differed across drugs (Chi-square test; $p = 0.012$). (C) Mean changes in the usAHP by each drug (one-way ANOVA followed by Holm–Sidak Multiple Comparison test). (D) Correlation between sensitivity of the usAHP to ouabain and the drop in the first R$_{in}$ measurement following stimulation. Traces illustrate R$_{in}$ responses for each example neuron at different ends of the correlation. (E) Same as (D) but for XE-991. (F) Correlation between sensitivity of the usAHP and the drop in R$_{in}$ after block of Kv7 by XE-991. The x-axis shows the ratio of the change in R$_{in}$ during the usAHP after and before XE-991. Thus, greater values reflect a greater block of the R$_{in}$ drop following XE-991. Inset for (E) shows the ability for XE-911 to counter the drop in R$_{in}$ in neurons where XE-991 also reduced the usAHP. The data underlying this figure can be found in S1 Data.

channels (30 μM Cd$^{2+}$), and Kv7 channels (10 μM XE-991) reduced the amplitude of the usAHP (Fig 2A). Ouabain and XE-991 reduced the usAHP by $\geq$10% in most neurons, while Cd$^{2+}$ had little-to-no effect (Fig 2B and 2C). These results suggest the dynamic Na$^+$ pump and Kv7 channels generate the usAHP, with no clear role for the activation of K$^+$ channels by Ca$^{2+}$ influx. Strikingly, variation in the sensitivity of the usAHP to ouabain and XE-991 was correlated with the activity-dependent drop in R$_{in}$. usAHPs with a large contribution from the dynamic Na$^+$ pump, which manifested as a greater sensitivity to ouabain, had the smallest reductions in R$_{in}$. Those with a lower pump contribution had larger changes in R$_{in}$ (Fig 2D). We observed the opposite trend for the Kv7 channel blocker, XE-991. Neurons with usAHPs containing a greater role of Kv7 channels had larger decreases in R$_{in}$ (Fig 2E). Moreover, the

greater the block of the usAHP by XE-991, the more XE-991 opposed the decrease in $R_{in}$ (Fig 2F). Thus, Kv7 channels contribute to both the usAHP as well as the accompanying change in $R_{in}$. This relationship did not exist for the block of $Ca^{2+}$ channels (S2 Fig). These results show that each feedback controller is scaled across neurons and suggest that cells with a greater role of the $Na^+$ pump have a smaller contribution from Kv7 channels and vice versa.

## Reciprocal function of dynamic $Na^+$ pump and Kv7 currents, but linear co-expression of mRNA transcripts

We next tested directly for reciprocal function of the dynamic $Na^+$ pump and Kv7 channels across neurons. For this, we measured the Kv7 component of the total outward current in voltage clamp and then estimated pump function through the sensitivity of the remaining usAHP to ouabain. Indeed, individual neurons with large Kv7 currents had a smaller block of the usAHP by 2 μM ouabain (Fig 3A and 3B$_1$). In contrast, neurons with relatively small Kv7 currents had greater apparent dynamic $Na^+$ pump function (Fig 3A-B$_2$). The relationship appeared to be hyperbolic, with a steeper slope through most of the distribution and a shallow slope at the top. Some cells exhibited a residual usAHP after the application of both ouabain and XE-991, suggesting that additional feedback processes exist in some neurons. Nevertheless, these results demonstrate an inverse relationship between the dynamic $Na^+$ pump and Kv7 current density, where graded combinations of these 2 feedback mechanisms integrate recent activity history of the neuron and then alter membrane excitability over the next minute.

The co-expression of ionic currents is often associated with correlated messenger RNA (mRNA) transcripts [11,25]. To understand the transcriptional basis of the relationship between the dynamic $Na^+$ pump and Kv7 channels, we assessed single-cell mRNA co-expression of genes that encode brain Kv7 channels (*KCNQ2*, *KCNQ3*, and *KCNQ5*) and the α3 subunit thought to give rise to dynamic activation of the $Na^+$ pump (*ATP1A3*) [21,26]. In contrast to the relationship at the functional level, mRNA copy numbers of *KCNQ2* and *KCNQ3* showed positive co-expression with *ATP1A3* across neurons (Fig 3C). We did not observe the same correlation for *KCNQ5* and *ATP1A3* (S3 Fig). In addition, 2 glutamate receptors did not correlate with *ATP1A3* (S3 Fig), supporting specificity of the mRNA correlation between *KCNQ2-KCNQ3* and the α3 subunit of the $Na^+$ pump. Thus, mRNA transcripts that encode Kv7 and the dynamic $Na^+$ pump are positively correlated across neurons. As the functions of these 2 proteins are inversely related, cellular processes that lie between mRNA and functional protein likely flip the relationship between these 2 mechanisms.

What proxy of activity does each feedback mechanism track? Loading of intracellular $Na^+$ during spiking activates the dynamic $Na^+$ pump, hyperpolarizing the membrane over tens of seconds as it exchanges 3 $Na^+$ for 2 $K^+$ ions [3]. This is consistent with our results showing that spiking causes the usAHP in cells with small $R_{in}$ changes and high sensitivity to 2 μM ouabain (Figs 1E$_1$ and 2D). For Kv7 channels, depolarization over several seconds can potentiate currents without spiking or $Ca^{2+}$ influx over the same time course [4]. Therefore, we tested if depolarization enhances the Kv7 component of the outward current (S4 Fig). For this, we measured the degree of $K^+$ current potentiation in the 25-second period following a 10-second step to −22 mV in voltage clamp, and then correlated it with the drop in $R_{in}$ during the usAHP measured in current clamp within the same neuron (S4A–S4C Fig). Neurons that reduced $R_{in}$ during the usAHP also had potentiated outward currents induced by depolarization, suggesting a relationship between these 2 processes (S4A Fig). XE-991 reduced potentiation of the outward current, indicating that Kv7 channels play a role in this response (S4D Fig). These results show that depolarization without spiking enhances Kv7 currents over a similar time course as spike-dependent feedback from the dynamic $Na^+$ pump.

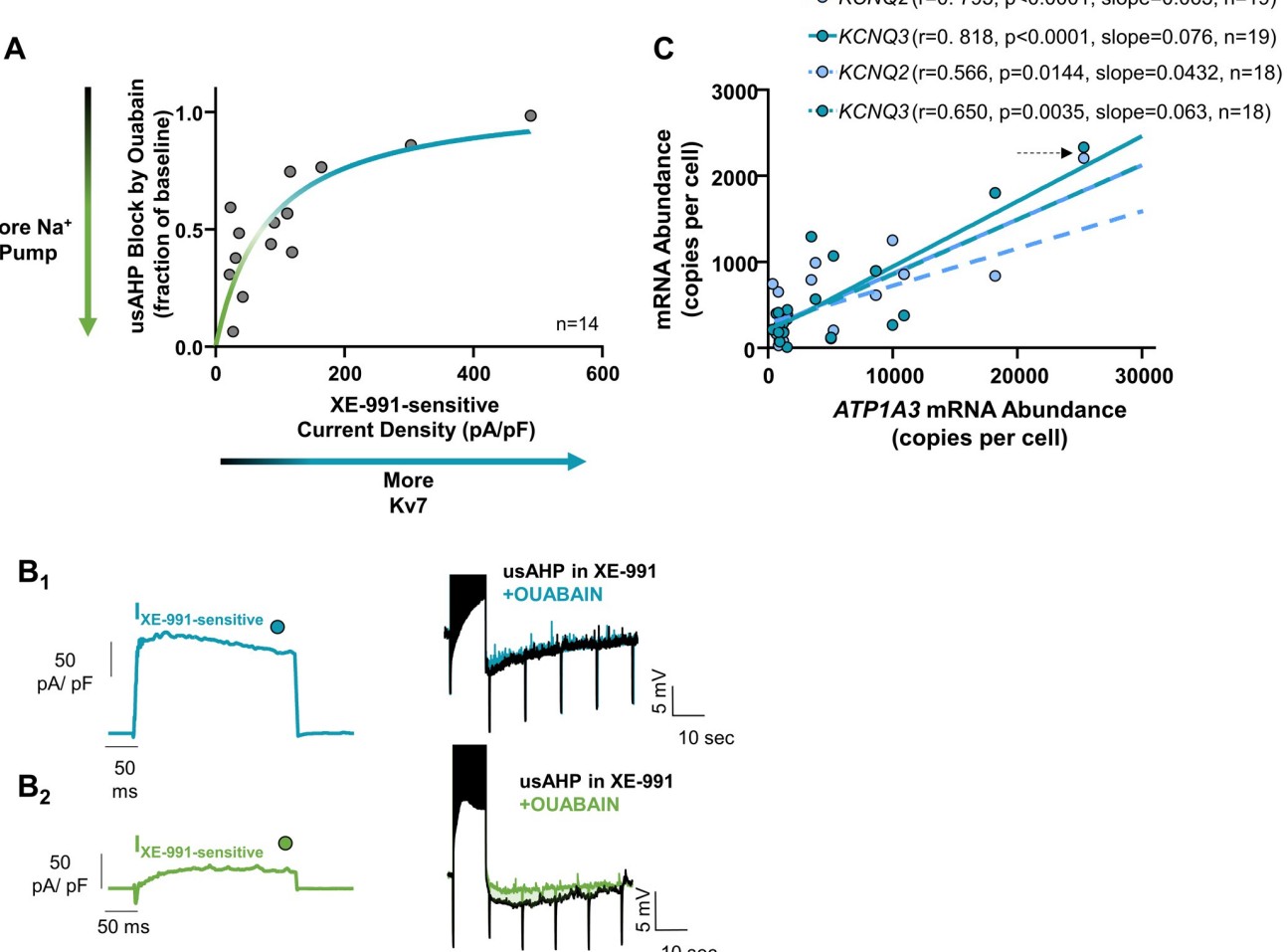

**Fig 3. Co-expression of Kv7 current density and the dynamic Na$^+$ pump at functional and mRNA levels.** (A) Inverse relationship between Kv7 current density and the dynamic Na$^+$ pump across neurons ($n$ = 14 neurons from $N$ = 10 animals); (B$_1$) shows a neuron with a relatively large Kv7 current density (XE-991 difference current recorded at −10 mV) and low function of the dynamic Na$^+$ pump (nearly no ouabain sensitivity of the remaining usAHP); (B$_2$) shows the opposite example, a neuron with relatively small Kv7 current density but a larger contribution of the dynamic Na$^+$ pump to the usAHP. (C) *KCNQ2* and *KCNQ3* (Kv7 subunits) mRNA expression from single neurons correlate with *ATP1A3*, the gene that codes for the Na$^+$ affinity α subunit of the Na$^+$ pump. Each point represents the mRNA for each gene from a single neuron ($n$ = 19 neurons from $N$ = 8 animals). Arrow points to values identified as outliers in the Grubbs' test. The dotted lines indicate the regression lines for each correlation run without the outlier point, with Pearson r, *p* value, and slope of the relationship shown in the key. The data underlying this figure can be found in S1 Data.

## Regulation of firing rate by combined feedback from dynamic Na$^+$ pump and Kv7 channels

We last sought to determine if respiratory motoneurons experience slow activity-dependent hyperpolarization in the intact network, and then, if reciprocal co-expression of the dynamic Na$^+$ pump and Kv7 channels plays a role in regulating firing rate during realistic activity. Fig 4A shows whole-cell current clamp recordings of identified vagal motoneurons from 3 different "semi-intact preparations" receiving physiological synaptic inputs from the respiratory rhythm generator, as described in reference [45]. After rhythmic bursts associated with lung ventilation, each neuron exhibited an afterhyperpolarization that lasted for several seconds before the next burst. Therefore, slow activity-dependent hyperpolarization occurs within the native network, suggesting a role for the dynamic Na$^+$ pump and Kv7 in regulating excitability to control respiratory motor behavior.

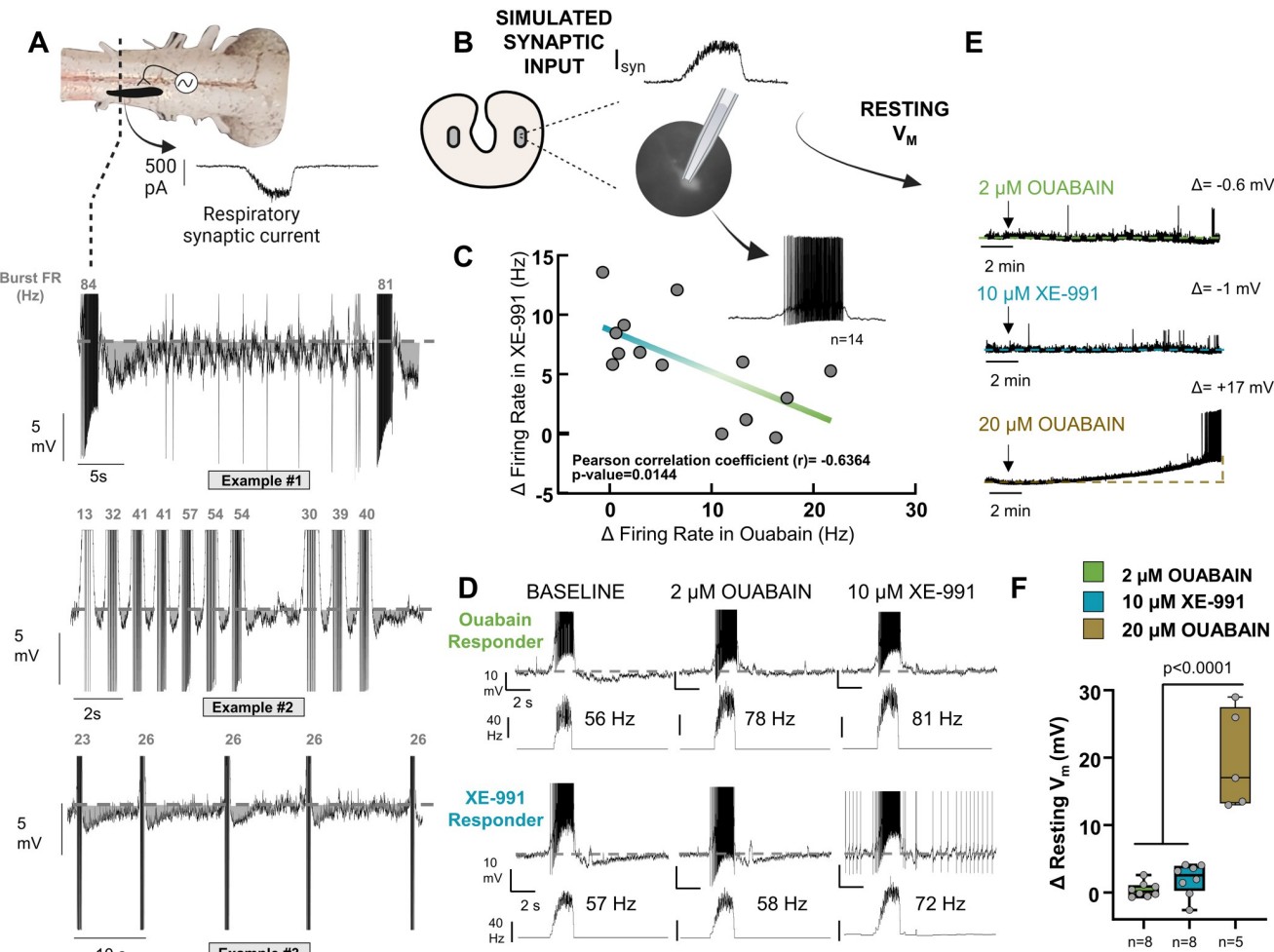

**Fig 4. The dynamic Na$^+$ pump and Kv7 channels contribute to regulation of firing rate through activity-dependent feedback.** (A) Identified vagal motoneurons receiving input from the respiratory rhythm generator exhibit slow afterhyperpolarization following rhythmic burst firing, demonstrating that physiological bursting is followed by prolonged membrane hyperpolarization, consistent with recruitment of the dynamic Na$^+$ pump and Kv7 channels. (B) Current clamp recordings of firing in response to stimulation of current input that mimics synaptic drive in slices. (C) Each neuron showed differential changes in firing rate in response to 2 μM ouabain and 10 μM XE-991 along a spectrum ($n = 14$ from $N = 8$ animals). (D) Top example shows a neuron with a large contribution of the Na$^+$ pump for the control of firing rate, with little input from Kv7 channels. Bottom example shows a neuron with little contribution from the pump but a large role for Kv7 in controlling firing rate. (E) Example recordings of resting membrane potential in silent neurons during application of 2 μM ouabain ($n = 8$ neurons from $N = 4$ animals), 10 μM XE-991 ($n = 8$ neurons from $N = 4$ animals), and 20 μM ouabain to block the "housekeeper" form of the Na$^+$ pump ($n = 5$ neurons from $N = 3$ animals). (F) Mean data for effects of each drug on resting membrane potential. The data underlying this figure can be found in S1 Data.

To determine the role of each feedback mechanism for the control of firing rate, we stimulated neurons in brain slices with current input that mimicked physiological synaptic drive (Fig 4B). This allowed us to simulate respiratory-related synaptic input in isolated motoneurons and avoid confounds associated with drug application in the entire network. Upon the sequential block of the dynamic Na$^+$ pump with 2 μM ouabain and then Kv7 with 10 μM XE-991, we observed a continuum of changes in firing rate during fictive synaptic input associated with breathing. Inhibition of the dynamic Na$^+$ pump led to progressive hyperexcitability in some neurons, and these neurons were less sensitive to the block of Kv7 channels (Fig 4C and 4D, top). In neurons that were less sensitive to the block of the dynamic Na$^+$ pump, hyperexcitability occurred in response to inhibition of Kv7 channels along a continuum (Fig 4C and

4D, bottom). These results indicate that reciprocally expressed sets of the dynamic Na$^+$ pump and Kv7 channels regulate firing rate during physiological bursting. To confirm the role of activity-dependent feedback, we applied each drug to silent neurons. Both blockers had little-to-no effect on the resting membrane potential (Fig 4E and 4F). Application of ouabain at 20 μM to inhibit the constitutively active Na$^+$ pump [20] led to depolarization and loss of the membrane potential, supporting 2 μM ouabain's actions on the "dynamic" form of the pump (Fig 4E and 4F). Little-to-no influence of these currents on the resting membrane potential supports the idea that dynamic chemical and physical signals during ongoing activity engage the Na$^+$ pump and Kv7. Thus, neurons within the same motor pool regulate their firing rates during bursting through unique combinations of activity-dependent feedback that track different cellular readouts of neuronal activity.

## Discussion

Neurons are thought to track the dynamics of a master command signal that follows firing rate to control their output. Many studies adhere to this interpretation for both short- and long-term regulation, stating it explicitly or implying the existence of a dominant signal that controls neuronal output over seconds to days [3,4,8,15,27–29]. Instead, we provide evidence that stable firing rates arise as neurons rapidly transduce distinct activity signals in varying combinations from cell to cell. These data reveal that consistent output from a population of neurons—ostensibly generating the same behavior—may emerge through multiple feedback systems with different relative weights across cells.

The major finding from this study is that individual neurons within the same population use unique combinations of molecular feedback controllers to maintain burst firing rates, with each process graded across the population. We attribute regulation to fast activity-dependent feedback because both the dynamic form of the Na$^+$ pump and Kv7 respond to neuronal activity and reduce excitability following strong bursting (Fig 2A and 2B) but have little contribution to the resting membrane potential (Fig 4F). Dynamic feedback from the Na$^+$ pump is thought to arise from the incorporation of an α3 subunit with a low affinity for intracellular Na$^+$, which serves to hyperpolarize the neuron as it extrudes Na$^+$ after firing [17]. Therefore, this form of the pump acts as both a spike sensor and effector for fast feedback regulation [3,17]. For Kv7 channels, membrane depolarization during activity—independent from spiking (Fig 1) and Ca$^{2+}$ influx (Fig 2)—rapidly potentiates currents to constrain neuronal output (S4 Fig). Membrane depolarization per se activates PI4 kinase to induce the production of PIP2 [30], which sensitizes Kv7 currents over a similar time course as we show here [4]. Thus, Kv7 channels seem to act as effectors that respond to fast voltage-sensitive signaling during bursting, potentially linked through the production of PIP2 or some other intermediate. These forms of feedback differ from classic examples of "homeostatic plasticity." That is, they use real-time dynamics of Na$^+$ and voltage to stabilize neuronal output, rather than engaging compensatory cell signaling that recovers activity following a strong perturbation. Nevertheless, different forms of homeostatic plasticity may also use multiple activity sensors and integration pathways within the same cell type [11,31,32]. Thus, we suggest that reciprocal scaling of multiple control mechanisms may be a feature for both rapid and long-term regulation of neuronal output.

How do neurons "know" to express each feedback system in their own unique way? Single-cell mRNA expression provides insight into transcriptional control that may give rise to this organization. Stable neuronal output can arise through variable expression of ionic conductances [33], but channels with similar properties tend to be inversely co-expressed across neurons [34]. This aligns with our results showing a reciprocal relationship between 2 feedback systems that act in the same direction, magnitude, and timescale. In contrast, mRNA transcripts

coding for Kv7 protein subunits (*KCNQ2* and *KCNQ3*) and the dynamic Na$^+$ pump (*ATP1A3*) are positively correlated across neurons. Given a reciprocal relationship at the functional level, positive mRNA correlations may seem contradictory. However, opposite relationships for mRNA abundance (positive) and corresponding channel currents (negative) have been demonstrated previously [11], and negative co-expression of channel transcripts has never been reported, even when the functional channel currents are inversely correlated [35]. Others have suggested the maintenance of channel mRNAs at roughly fixed ratios, which manifest as positive correlations across neurons, may reflect critical gene modules needed to generate certain neuronal properties regardless of the final relationship of the functional currents [35]. Why neuronal populations appear to maintain mRNAs only at linear positive relationships, and what mechanisms flip the direction of these relationships along the path from mRNA to protein function such as we present, is not known. Nevertheless, positive correlations of channel mRNAs are thought to be actively maintained largely through slow feedback from voltage-dependent signaling in rhythmic motoneurons [36]. Thus, we speculate that activity-dependent transcription constrains relevant mRNA relationships, and then through undefined cellular mechanisms, guides the proper combination of these rapid feedback systems in each cell.

Each neuron in the vagal motor pool appears to continuously sense different aspects of its activity to constrain population motor output for breathing. What is the advantage of multiple rapid feedback systems reciprocally balanced across neurons? Many behaviors require stable neuronal output. A failure to maintain stability can lead to network dysfunction if left unchecked, and fast feedback as we describe here may allow neurons to correct course nearly in real time to avoid disease states [9,37]. We suggest that scattering 2 molecular controllers across neurons may prevent the complete loss of stability at the population level during transient disturbances to 1 feedback system. For example, activity of the Na$^+$ pump is tied to local ATP concentrations [38], and the high consumption of ATP used to fuel other costly neuronal processes may limit the ability of the pump to stabilize firing rate. Therefore, behaviors that involve energetically expensive synapses [39] or activity during severe metabolic stress [40] may spiral out of control if feedback in every neuron relied solely on the dynamic Na$^+$ pump. Indeed, mutations in the dynamic form of the Na$^+$ pump and Kv7 channels are implicated in epilepsy [41,42], where states of hyperexcitability are commonly associated with mitochondrial dysfunction [43]. Thus, although we uncovered this feedback organization in the amphibian brainstem, we suggest that it may extend to other species, whereby changes in the balance of these 2 feedback systems could lead to disordered circuit function.

A grand challenge for modern neuroscience is to understand how neurons maintain stable function for many days, weeks, and years [35]. As neurological disease occurs when firing rate strays from a healthy range, regulatory principles have broad implications for the treatment of vast brain disorders. Our results highlight that animals may scale different combinations of regulatory mechanisms throughout a population of neurons, which presents a challenge for defining uniform principles and applying them to disease. Neuronal stability likely involves a suite of interwoven mechanisms acting over long and short timescales through activity-dependent and neuromodulatory feedback, as well as genetic control [36,44]. Data presented here emphasize that overarching frameworks for the control of neuronal output may need to account for assortments of feedback processes spread variably across neurons of the same population that ultimately give rise to stable behaviors.

## Materials and methods

All experiments performed were approved by the Institutional Animal Care and Use Committee (IACUC) at the University of North Carolina at Greensboro (#19–006 and #2022–1163).

Adult American Bullfrogs, *Lithobates catesbeianus*, were purchased from Rana Ranch (Twin Falls, Idaho) and were housed in 20-gallon plastic tanks containing dechlorinated tap water at room temperature bubbled with room air. Tanks were bubbled continuously with room air and access to both wet and dry areas was available. Frogs were maintained on a 12-hour light/ dark cycle and fed once per week. Water was cleaned daily and changed as needed. All experiments occurred during the light cycle.

### Preparation of brainstem slices and labeling of respiratory motoneurons

Brain slice preparations containing labeled respiratory motoneurons that innervate the glottal dilator (a respiratory muscle in frogs) were generated using standard methods [23]. Briefly, frogs were anesthetized with 1 ml isoflurane in an approximately 1 L container until loss of the toe-pinch reflex. Euthanasia was achieved via rapid decapitation. Brainstem dissection was performed in ice-cold artificial cerebrospinal fluid (aCSF) bubbled with 98.5% $O_2$/1.5% $CO_2$ where the forebrain, optic tectum, brainstem, and spinal cord were removed from the head. aCSF was composed of (in mM): 104 NaCl, 4 KCl, 1.4 $MgCl_2$, 7.5 Glucose, 1 $NaHPO_4$, 40 $NaHCO_3$, 2.5 $CaCl_2$. The forebrain was quickly crushed, the spinal cord was trimmed caudal to spinal nerve III, and the brainstem was transferred to an approximately 6 ml Sylgard-coated Petri dish, superfused with aCSF using a peristaltic pump (Watson-Marlow, Wilmington, Massachusetts, USA). The brainstem preparation was stabilized by the addition of 2 pins, 1 in the rostral midbrain and 1 in the spinal cord. We then loaded the fourth branch of vagus nerve with a fluorescent dye (Dextran, Tetramethylrhodamine, 3000 MW, anionic) for 1 hour on 1 side and 2 hours on the other (3 hours total), as approximately 75% of these neurons receive respiratory-related synaptic input associated with breathing [45]. Transverse 300 µm-thick slices of the brainstem (for brainstem slices) or the dorsal portion of the brainstem (for the "semi-intact" network preparation) [45] were then cut using a Vibratome Series 1000 sectioning system in ice-cold aCSF bubbled with 98.5% $O_2$/1.5% $CO_2$. Tissue preparations were given 1 hour to recover and maintained in room temperature aCSF bubbled with 98.5% $O_2$/1.5% $CO_2$ throughout the experiments.

### Drug solutions

All drugs were dissolved in standard aCSF. Cadmium ($Cd^{2+}$; Thermo Fisher Scientific, Waltham, Massachusetts, USA) at 30 µM was used to reduce the function of voltage-gated $Ca^{2+}$ channels [46]. TTX citrate at 250 nM (HelloBio, Princeton, New Jersey, USA) was used to block voltage-gated $Na^+$ channels. Ouabain (2 µM or 20 µM; HelloBio, Princeton, New Jersey, USA) was used to block the $Na^+$-$K^+$ ATPase. Low-dose ouabain is relatively selective for the dynamic form of the pump, as the α3 subunit has a higher affinity for ouabain compared to the constitutively active isoform [20]. XE-991 dihydrochloride (HelloBio, Princeton, New Jersey, USA) at 10 µM was used to block Kv7 channels.

### Electrophysiology

**Whole-cell current clamp.** Experiments using current clamp electrophysiology were performed on the Axopatch 200B amplifier or the Multiclamp 700A amplifier and digitized with a Digidata 1550B (Molecular Devices, San Jose, California, United States of America). Patch pipettes for current clamp recordings were pulled on a P-87 Puller (Sutter Instruments, Navato, California, USA) with a tip resistance between 2.5 and 5 MΩ when filled with intracellular solution containing (in mM): 110 K-gluconate, 2 $MgCl_2$, 10 HEPES, 1 $Na_2ATP$, 0.1 $Na_2GTP$, 2.5 EGTA, pH = 7.2 with KOH. Glass used for patch pipettes was purchased from either Harvard Apparatus (1.5 ODx1.17IDx100L mm: Holliston, MA, USA) or World

Precision Instruments (1.5ODx1.12ID: Sarasota, FL, USA). Cells that were positive for fluorescence (excitation, 540 nm: emission, 605 nm) were visualized at 40× and then approached with the patch pipette using IR-DIC optics. After breaking the GΩ seal, cells were given between 2 and 3 minutes to stabilize. Cells with obvious health issues (e.g., depolarized membrane potentials, no action potentials) were discarded. Membrane potential of all neurons were held at −55 mV in current clamp mode by injecting a DC bias current. Neurons were initially injected with a range of currents to estimate maximal sustained firing frequency. If cells entered depolarization block during current injection, current was reduced to obtain repetitive firing throughout the 10 second pulse. Maximum current injected was 1 nA in this study. Following the 10 second stimulation period, we observed the usAHP over the next 75 seconds. Input resistance ($R_{in}$) was measured immediately before stimulation, and then following the 10 second stimulation, every 10 seconds after during the 75 second recovery period. Input resistance was measured by delivering a −100 pA current injection for 200 ms. Following the initial usAHP measurement, a number of neurons were exposed to 2 μM ouabain, 10 μM XE-991, and 30 μM $Cd^{2+}$ for approximately 15 minutes. In experiments that assessed the role of spiking on the usAHP, 250 nM TTX was included in the aCSF or stimulation was delivered to reach depolarization block. Action potentials are truncated in all traces to highlight the usAHP.

To determine whether slow hyperpolarizations follow burst firing in the intact respiratory network, we used a semi-intact preparation that allows simultaneous access to extracellular nerve root recordings and motoneuron patch clamp recordings during the ongoing respiratory rhythm [45]. We also used this preparation to make recordings of synaptic currents onto the motoneuron that are associated with breathing. We then took this synaptic current recording and converted it into a current clamp protocol. This allowed us to deliver current to the cell that simulated synaptic input close to what vagal motoneurons would normally receive during respiratory bursts, but in neurons isolated in brainstem slices (Fig 4B). For this, we constructed a current clamp protocol that injected cells with current that matched the exact waveform of synaptic input recorded from the intact network and paced them at a frequency consistent with the respiratory rhythm. Thus, we could restore breathing-related motor patterns in neurons that were disconnected from the rhythmic network [22,45]. The protocol consisted of 10, approximately 1-second injections of physiological synaptic input separated by an approximately 9-second-long interval.

For current clamp experiments assessing firing rate in response to physiological stimulation, cells were allowed the same recovery period after whole-cell access as previous experiments. Cells were held at approximately −55 mV in current clamp mode. Only neurons that could be maintained at approximately −55 mV by the bias current were included in these experiments. The baseline measurement of burst firing was taken 5 minutes after the cell was stimulated with "fictive" respiratory-related synaptic drive. After the baseline measurement, 2 μM ouabain was washed in for 15 minutes followed by 10 μM XE991 mixed with 2 μM ouabain for the next 15 minutes. Fictive synaptic input associated with breathing was continuously delivered during all drug exposures. This same drug exposure protocol was given to neurons that did not receive rhythmic input to assess their impact on resting $V_m$. $V_m$ measurements represent the average for 1 minute before and after 15 minutes of each drug treatment.

**Whole-cell voltage clamp.** All voltage clamp experiments were performed using the Axopatch 200B amplifier. Patch pipettes were pulled using the same puller as the current clamp experiments with a tip resistance between 2.5 and 4 MΩ using the same glass as the current clamp experiments. Tips were wrapped in parafilm to reduce pipette capacitance and filled with standard intracellular solution. Cells were given 1 minute to recover following establishment of whole-cell access. Series resistance was compensated between 75% and 98% and only

cells with an effective series resistance of 3 MΩ or lower with a change of less than 0.5 MΩ were used in this study. The outward K$^+$ current was then evoked with a step protocol, using a −80 mV holding potential to +0 mV in 10 mV increments for 250 ms every 500 ms. After the initial outward K$^+$ current was determined, XE991 (10 μM) was washed in for approximately 7 minutes, and the step protocol was repeated. The Kv7 channel current density at −10 mV was determined by taking the difference between the steady-state outward K$^+$ current in aCSF and after wash-in of XE991, normalized for the cell capacitance read off the "whole-cell capacitance" dial on the amplifier. Kv7 is typically a slow activating and non-inactivating current in neurons. However, in some neurons, we observed slow inactivation of the XE-991 difference current. To enhance our confidence that we assessed the density of the "true" Kv7 component, we only report data at the end of the 500 ms step. Current traces were filtered offline with a lowpass Boxcar filter.

To assess of outward current potentiation in response to brief depolarization, we obtained a baseline outward current measurement by stepping from −80 to −10 mV for 500 ms. We then stepped to −22 mV for 10 seconds. Neurons were then stepped back to −80 for 5 seconds and then stepped to −10 mV for 500 ms every 10 seconds to assess current amplitude following depolarization to −22 mV. During the step to −22 mV, no cells were included showed evidence of unclamped action potentials throughout the 10-second step, and those that did were discarded. In a subset of neurons with potentiation following depolarization, we applied 10 μM XE-991 and then reran the protocol to assess the contribution of Kv7 currents.

## Single-cell cDNA synthesis and quantitative PCR

To determine mRNA expression for Kv7 and Na$^+$ pump subunits, we used single-cell quantitative PCR. Briefly, after making a whole-cell recording with 2.5 μL of standard pipette filling solution in the tip, gentle and progressively increasing negative pressure was applied through a 60 ml syringe connected the pipette holder. Pressure began gentle, as to not disrupt the seal between the pipette and the cell membrane over the first 4 minutes, and then suction was increased over the next 2 minutes. During the suction processes, the membrane potential was stepped from −5 mV to +20 mV at 5 ms intervals in voltage clamp to hold RNA in the pipette tip solution [47]. We then transferred the contents of the pipette into 100 μL of lysis buffer and isolated total RNA using a column-based extraction (Quick-RNA MicroPrep Kit, Zymo Research). We synthesized cDNA from each single-cell RNA sample using SuperScript IV VILO in a 14 μL reaction according to the manufacturer's instructions (Thermo Fisher Scientific, Waltham, Massachusetts, USA). We then preamplified our targets of interest from each cell's 14 uL cDNA reaction (*KCNQ2*, *KCNQ3*, *KCNQ5*, *ATP1A3*, *GluR1*, *GluR2*) using PerfeCTa PreAmp Supermix (Quanta BioDesign, Plan City, Ohio) in a 20 μL reaction according to the manufacturer's instructions. After preamplification, the contents of the tube were diluted 7.5-fold in nuclease-free water (150 μL final volume), which was then used as the template for real-time quantitative PCR.

PCR primers for *KCNQ3*, *KCNQ5*, *ATP1A3*, *GluR1*, and *GluR2* were designed based on sequences identified in the coding DNA sequence for *Lithobates catesbeianus*. Briefly, we used annotated amino acid sequences for our targets of interest from *Rana temporaria* as a query in the *Lithobates catesbeianus* amino acid database. This produced hits with high amino acid sequence conservation. We then performed a reciprocal BLAST against the entire nonredundant protein database using hits from *Lithobates catesbeianus* to verify the identity of the target. Accession numbers were then used to identify the open reading frame in the CDS to design PCR primers. We could not identify *KCNQ2* sequences in the bullfrog CDS, likely due to low coverage of the *Lithobates catesbeansus* draft genome [48]. Therefore, we found the regions of the *KCNQ2* coding sequence in *Rana temporaria* (a species closely related to *L.*

**Table 1. Primer sequences for SYBR Green qPCR assays.**

| Target | Forward | Reverse | Efficiency |
|--------|---------|---------|------------|
| *KCNQ2* | AGAGGAGGGACGTGGAAACT | GAGGCAGAGGAAGCCAATGT | 98.088% |
| *KCNQ3* | CGGGTTCAGCATTTCCCAAG | TCGGTGTCCGTTTCACCTTC | 85.465% |
| *KCNQ5* | AGCTCCACCCCCATTGAAAC | TTTGAGGCAACAAGGCAGGA | 84.414% |
| *ATP1A3* | GCGCAAGATTGTGGAATTCACT | TGATGACATCGGCCCATTGTAC | 104 |
| *18S rRNA* | CAGGCCGGTCGCCTGAATAC | GGCCCCAGTTCCGAAAACCA | 101.068% |

*catesbeianus*) and *N. parkeri* (a more distantly related frog species) with high similarity to design PCR primers for *KCNQ2*. Our rationale was that close sequence identity at the nucleotide level between these 2 species would allow us to design primers for use in *Lithobates catesbeianus*.

Primer sets for all targets were each validated in-house with a series of four 4-fold dilutions of brainstem cDNA using the thermal cycling conditions stated below. All primer sets used here produced efficiencies greater than 80% (Tables 1 and 2) and a single peak in the melt curve, suggesting the amplification of 1 PCR product. In several cases during primer validation, primer sets that produced 1 peak in the melt curve with cDNA from bulk brain tissue showed multiple peaks in the melt curve after single cell samples were preamplified. These primer sets were discarded and redesigned. Therefore, each primer set used in this study was validated on both bulk tissue and single cell samples following cDNA synthesis and preamplification.

All neurons in this experiment were assessed for the expression of each target gene. Quantitative PCR was run in 10 μL reactions containing 2.5 μM forward and reverse primers following the instructions of the 2X SYBR Green Mastermix (Applied Biosystems, Thermo Fisher Scientific, Waltham, Massachusetts, USA). Assays were run on 96-well plates on an Applied Biosystems QuantStudio 3 (Applied Biosystems, Thermo Fisher Scientific, Waltham, Massachusetts, USA) using the following cycling conditions according to the SYBR Green instructions: 50 °C-2m, 95 °C-10 m, 95 °C-15 s, 60 C-1 m. Following 40 cycles of PCR (95 °C-15 s, 60 °C-1 m), melt curves for all PCR products were acquired by increasing the temperature in increments of 0.3 °C for 5 seconds from 60 °C to 95 °C. GluR1 and GluR2 were run using duplexed probed-based assays. For this, we used the same primer concentration as described for SYBR Green assays, 312.5 nM reporter probes, and followed the instructions of the 5× PerfeCTa qPCR Toughmix mastermix (Quanta Bio); 18S ribosomal RNA was run to ensure quality of the sample. Absolute quantitation of transcript abundance was estimated through copy number standard curves as previously described [36].

## Statistics analysis

Statistics were performed in GraphPad 9.0.0 (GraphPad Software, San Diego, California, USA). Correlations were determined using the Pearson correlation coefficients. To determine

**Table 2. Primer sequences for Probe-based qPCR assays.**

| Target | Forward primer | Reverse primer | Efficiency |
|--------|----------------|----------------|------------|
| *GluR1* | GGGATGAACAGAGTGAGAAAGTC | CCTCCCACCTTCATGGTGTC | 97.5% |
| *GluR2* | CCTCCCACCTTCATGGTGTC | GCCAAGTCCTCCAACAAGAATG | 94.8% |
| | **Prober sequences** | **Probe-quencher** | |
| *GluR1* | ATGCATATCTCTTGGAGTCCACGATGA | FAM-BHQ1 | |
| *GluR2* | TGCACTCAGTCTGAGTAATGTGGC | Texas Red-BHQ2 | |

significance between group means, we used unpaired $t$ tests. When comparing "before and after" responses on the same neuron, we used paired $t$ tests. One-way ANOVA was used to compare datasets with 1 main effect, and two-way ANOVA was used to analyze datasets with 2 main effects. Following the one-way ANOVA, Holm Sidak's multiple comparison test was used to assess pairwise comparisons. Proportions of occurrences were analyzed using a Chi-squared test. Significance was accepted when $p < 0.05$.

## Supporting information

**S1 Fig. Absolute changes in $R_{in}$ following activity.** Data shown here are the same as shown in Fig 1C, but instead plotted as absolute rather than relative $R_{in}$. Red represents the smallest 1/3 of $R_{in}$ changes in the dataset, green represents the middle 1/3, and blue shows the largest $R_{in}$ drops in the dataset. The data underlying this figure can be found in S1 Data.
(TIF)

**S2 Fig. No relationship between sensitivity of the usAHP to $Cd^{2+}$ and the block of the drop in $R_{in}$ by $Cd^{2+}$ ($n = 14$).** The data underlying this figure can be found in S1 Data.
(TIF)

**S3 Fig. *KCNQ5* and 2 AMPA-glutamate receptors (*GluR1* and *GluR2*) do not correlate with the *ATP1A3* mRNA expression.** Each data point represents the mRNA expression from a single neuron ($n = 19$). The data underlying this figure can be found in S1 Data.
(TIF)

**S4 Fig. Brief depolarization potentiates Kv7 currents over the timescale of the usAHP.** (A) Neurons with $R_{in}$ decreases during the usAHP also have outward $K^+$ currents potentiated by brief depolarization. The box highlights examples where $R_{in}$ was reduced during the usAHP and outward currents were potentiated by 10 seconds of depolarization ($n = 38$); (B) shows the protocol to assess outward current potentiation and the activity-dependent change in $R_{in}$ during the usAHP in the same neuron. The circle indicates that data are reported at the end of the step. (C) Example traces showing potentiation of the outward current by depolarization to −22 mV, which decays following stimulation over the following 10 seconds of seconds. The green trace shows the same neuron, but after the application of XE-991. XE-991 reduced the potentiation, demonstrating a role for Kv7 channels. (D) Individual data points from 4 experiments before and after application of XE-991. Each cell had potentiated outward currents that were reduced by XE-991 ($n = 4$). Two-way ANOVA ($p = 0.0216$; Drug × time interaction). The data underlying this figure can be found in S1 Data.
(TIF)

**S1 Data. Raw numerical data.**
(XLSX)

## Author Contributions

**Conceptualization:** Joseph M. Santin.

**Data curation:** Sarah Pellizzari, Min Hu, Lara Amaral-Silva, Sandy E. Saunders.

**Formal analysis:** Sarah Pellizzari, Joseph M. Santin.

**Funding acquisition:** Joseph M. Santin.

**Investigation:** Sarah Pellizzari, Joseph M. Santin.

**Methodology:** Sarah Pellizzari, Min Hu, Lara Amaral-Silva, Sandy E. Saunders, Joseph M. Santin.

**Supervision:** Joseph M. Santin.

**Validation:** Sarah Pellizzari, Min Hu, Joseph M. Santin.

**Visualization:** Joseph M. Santin.

**Writing – original draft:** Sarah Pellizzari, Joseph M. Santin.

**Writing – review & editing:** Sarah Pellizzari, Min Hu, Lara Amaral-Silva, Sandy E. Saunders, Joseph M. Santin.

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
