## [Editor Report · Decision Letter 0]

5 Oct 2022

*Dear Dr Santin, 

Thank you for submitting your revamped manuscript entitled "Neuron populations use variable combinations of short-term feedback mechanisms to stabilize firing rate" for consideration as a Short Reports by PLOS Biology.

As we discussed before, we are interested in sending your submission out for external peer review. Before we can send your manuscript to reviewers though, we will need you to complete your submission by providing the metadata that is required for full assessment. To this end, please login to Editorial Manager where you will find the paper in the 'Submissions Needing Revisions' folder on your homepage. Please click 'Revise Submission' from the Action Links and complete all additional questions in the submission questionnaire.

Once your full submission is complete, your paper will undergo a series of checks in preparation for peer review. After your manuscript has passed the checks it will be sent out for review. To provide the metadata for your submission, please Login to Editorial Manager (https://www.editorialmanager.com/pbiology) within two working days, i.e. by Oct 07 2022 11:59PM.

Kind regards,

Kris

Kris Dickson, Ph.D. (she/her)

Neurosciences Senior Editor/Section Manager

PLOS Biology

kdickson@plos.org

---

## [Decision Letter · Decision Letter 1]

10 Nov 2022

Dear Dr Santin,

Thank you for your patience while your manuscript "Neuron populations use variable combinations of short-term feedback mechanisms to stabilize firing rate" was peer-reviewed at PLOS Biology. It has now been evaluated by the PLOS Biology editors, an Academic Editor with relevant expertise, and by 2 independent reviewers. 

In discussing the reviewers' feedback with our Academic Editor, our Academic Editor argued that the insights from this work are potentially significant enough for PLOS Biology if the technical issues can be addressed. Therefore, we would like to invite you to revise the work to thoroughly address the reviewers' technical concerns. You will find their detailed feedback at the end of this email. We expect to receive your revised manuscript within 3 months. Please email us (plosbiology@plos.org) if you have any questions or concerns, or would like to request an extension. 

**IMPORTANT - SUBMITTING YOUR REVISION**

*Re-submission Checklist*

*Published Peer Review*

*PLOS Data Policy*

*Blot and Gel Data Policy*

Sincerely,

Kris

Kris Dickson, Ph.D., (she/her)

Neurosciences Senior Editor/Section Manager

PLOS Biology

kdickson@plos.org

REVIEWS:

Do you want your identity to be public for this peer review?

Reviewer #1: No

Reviewer #2: No

Reviewer #1: Overview: This manuscript addresses an important and fundamental problem: how neurons maintain their characteristic and fundamental patterns of electrical excitability. This manuscript presents some novel results that are intriguing, and argue for the first time, that a population of neurons uses two different cellular mechanisms (a Na pump and a K channel), in different ratios, to achieve their characteristic activity patterns. 

While the work has been cleverly and carefully done, I am always concerned by the extent to which the findings depend on the specificity of pharmacological agents. How do we know that the drugs used are specific and used at appropriate concentrations?

The Discussion is quite thoughtful. 

Specific comments:

1. line 34. I don't think that the reference to Fig, 1A is needed here, and I don't like referring to figures in the Introduction.

2. Figure 1F. the format of these plots is a bit strange…the lines with no points make it look like there is a continuous function while there are just two points connected? Can you plot differently?

3. Figure 1C, I understand why these data were normalized, but isn't there a way to plot the data that doesn't lose the variance of the data by not normalizing? Likewise Fig 1D?

4. What evidence is there that XE-991 is specific for Kv7 channels? In this species?

Reviewer #1 Cross Comments:

I couldn't get back to the original manuscript, so I can't see what references the other reviewer was arguing about. More critically, I sort of doubt that this idea was already in the literature, and I don't understand the other reviewer's minor comment #3. I still think the paper is quite intriguing and original and think you should give the authors the chance to revise it.

Reviewer #2: This interesting manuscript studies regulation of the breathing related activity of vagal motoneurons in an amphibian model. The goal was to evaluate the role of activity-dependent regulation of network bursting activity by the Na+/K+ pump and voltage-gated K+ channels (Kv7). The key finding is that a given vagal motoneuron has differing levels of pump activity and Kv7 channels. The methods are straightforward, the paper is mostly well written, and the results support the concept of differential molecular controllers of membrane excitability.

Major concerns.

There are some important weaknesses. First, the idea is not new. More exhaustive work on this problem has been published based on the recording of both lumbar motoneurons (your reference 17) and preBotzinger complex inspiratory interneurons (Krey et al., Front Neural Circuits. 2010 Nov 29; 4:124), which is not cited. Second, almost all results are based on correlations, with many of the relations relying on relatively small sample sizes. For example, the data in Fig. 3C seems to be heavily influenced by two data points in the upper right quadrant of the graph; what is the correlation coefficient without these two data points? I do understand that short reports do not require a full data set, but novelty needs to be made crystal clear. Third, the paper lacks discussion of how this mechanism would work to control the network's cycle period to affect an increase or decrease in ventilatory output. 

Minor Concerns.

1. The mRNA results are confusing and insufficiently discussed and interpreted.

2. Lines 148-150. The figures appear to be mis-labeled.

3. The simulated synaptic input studies are a nice addition, though it should be mentioned that even though this may be better than bath-application of drugs (Line 143), changes in voltage-dependent ion channels cannot be avoided.

Reviewer 2 Cross Comments:

I remain unconvinced that the idea is brand new but asked an extremely bright research Asst. Professor in the lab to also have a look as well. Their comments:

I found the manuscript interesting and the experiments were carefully designed, however I don’t think that the results are significant enough to be published in PLOS Biology short reports.

The novelty of these results is definitely questionable. There are many publications citing multiple mechanisms within one neuron type that underly slow afterhyperpolarization. The one potentially novel piece is the finding that motoneurons have graded expression of the two mechanisms described, i.e if Na+ pump is high then Kv7 is low and vice versa, but that seems intuitive given the remarks in the introduction that neurons have a “dominant” sensory signal for a given regulated variable. In addition, as written, the authors seem to suggest that Na+ pump and Kv7 are the only two mechanisms by which these neurons maintain their firing rates. I don’t feel like they performed a thorough investigation of other possible mechanisms and, therefore, the idea that the gradation of these two specific channels is the sole mechanisms is overstated. Finally, the PCR data doesn’t add value. In my opinion, imunohistochemistry showing gradation of these two channels across this neuron population would have been better to support for their ephys data and would make the manuscript stronger, but without this, what is left is a lovely, yet not groundbreaking, ephys data set.

---

## [Decision Letter · Decision Letter 2]

16 Dec 2022

Dear Joseph,

Thank you for your patience while we considered your revised manuscript "Neuron populations use variable combinations of short-term feedback mechanisms to stabilize firing rate" for publication as a Short Reports at PLOS Biology. This revised version of your manuscript has been evaluated by the PLOS Biology editors, the Academic Editor and the original reviewers.

I am happy to say that both reviewers were impressed with your revisions, so we are likely to accept this manuscript for publication. At this point we simply need you to modify your figure legends to include information on where the underlying data can be found (e.g. “The underlying data supporting Fig X, panel Y can be found in file Z.”).

As you address this request, please take this last chance to review your reference list to ensure that it is complete and correct. If you have cited papers that have been retracted, please include the rationale for doing so in the manuscript text, or remove these references and replace them with relevant current references. Any changes to the reference list should be mentioned in the cover letter that accompanies your revised manuscript.

We expect to receive your revised manuscript the first week of 2023. (Please note that I will be away from Dec 19-Jan 2, so there is no rush to get the revision in before the beginning of January.)

*Published Peer Review History*

*Press*

Sincerely,

Kris

Kris Dickson, Ph.D., (she/her)

Neurosciences Senior Editor/Section Manager,

kdickson@plos.org,

PLOS Biology

Reviewer remarks:

Do you want your identity to be public for this peer review?

Reviewer #1: No

Reviewer #2: Yes: Ralph Fregosi

***Reviewer #1: I think the authors have been responsive to these reviews

***Reviewer #2: The authors have made a Herculean effort to revise the manuscript, and to respond to the critiques. I feel that the paper is now suitable for publication.

---

## [Editor Report · Decision Letter 3]

19 Dec 2022

Dear Dr Santin,

Thank you for the submission of your revised Short Reports "Neuron populations use variable combinations of short-term feedback mechanisms to stabilize firing rate" for publication in PLOS Biology. On behalf of my colleagues and the Academic Editor, Tom Südhof, I am pleased to say that we can in principle accept your manuscript for publication, provided you address any remaining formatting and reporting issues. These will be detailed in an email you should receive within 2-3 business days from our colleagues in the journal operations team; no action is required from you until then. (Note - given the upcoming holidays and office closures, it may take longer than the standard 2-3 days for the operations team to reach out.) Please note that we will not be able to formally accept your manuscript and schedule it for publication until you have completed any requested changes they ask for.

PRESS

We frequently collaborate with press offices. If your institution or institutions have a press office, please notify them about your upcoming paper at this point, to enable them to help maximize its impact. If the press office is planning to promote your findings, we would be grateful if they could coordinate with biologypress@plos.org. If you have previously opted in to the early version process, we ask that you notify us immediately of any press plans so that we may opt out on your behalf.

Sincerely, 

Kris

Kris Dickson, Ph.D., (she/her)

Neurosciences Senior Editor/Section Manager

PLOS Biology

kdickson@plos.org